Kombucha: a novel model system for cooperation and conflict in a complex multi-species microbial ecosystem

May Alexander 1 2
Narayanan Shrinath 3
http://orcid.org/0000-0003-0807-164X Alcock Joe 4
http://orcid.org/0000-0003-4111-2415 Varsani Arvind 1 5 6 7
Maley Carlo 1 3
http://orcid.org/0000-0002-7128-670X Aktipis Athena 2 3 5 7 aktipis@asu.edu
1 School of Life Sciences, Arizona State University , Tempe, AZ , USA
2 Department of Psychology, Arizona State University , Tempe, AZ , USA
3 The Biodesign Center for Biocomputing, Security and Society, Arizona State University , Tempe, AZ , USA
4 University of New Mexico , Albuquerque, NM , USA
5 The Biodesign Center for Fundamental and Applied Microbiomics, Center for Evolution and Medicine, School of Life Sciences, Arizona State University , Tempe, AZ , USA
6 Structural Biology Research Unit, Department of Clinical Laboratory Sciences, University of Cape Town , Cape Town , South Africa
7 Center for Evolution and Medicine, Arizona State University , Tempe, AZ , USA
Gillespie Joseph
Electronic publication date: 2019 Sep 3
Publication date: 2019
Volume: 7
Electronic Location ID: e7565
Received 2019 Mar 13; Accepted 2019 Jul 29
Copyright: © 2019 May et al.
Copyright year: 2019
Copyright holder: May et al.
License: This is an open access article distributed under the terms of the Creative Commons Attribution License, which permits unrestricted use, distribution, reproduction and adaptation in any medium and for any purpose provided that it is properly attributed. For attribution, the original author(s), title, publication source (PeerJ) and either DOI or URL of the article must be cited.
License URL: https://creativecommons.org/licenses/by/4.0/

Keywords: Fermentation, Cooperation, Conflict, Competition, Symbiosis, Community, Biofilm, Microbiome, Artificial selection, Evolution

Funding: National Cancer Institute of the National Institutes of Health U54CA217376 John Templeton Foundation 46724 Research reported in this article was supported by the National Cancer Institute of the National Institutes of Health under Award Number U54CA217376 and John Templeton Foundation Grant Number 46724. The funders had no role in study design, data collection and analysis, decision to publish, or preparation of the manuscript.

==============================
Kombucha, a fermented tea beverage with an acidic and effervescent taste, is composed of a multispecies microbial ecosystem with complex interactions that are characterized by both cooperation and conflict. In kombucha, a complex community of bacteria and yeast initiates the fermentation of a starter tea (usually black or green tea with sugar), producing a biofilm that covers the liquid over several weeks. This happens through several fermentative phases that are characterized by cooperation and competition among the microbes within the kombucha solution. Yeast produce invertase as a public good that enables both yeast and bacteria to metabolize sugars. Bacteria produce a surface biofilm which may act as a public good providing protection from invaders, storage for resources, and greater access to oxygen for microbes embedded within it. The ethanol and acid produced during the fermentative process (by yeast and bacteria, respectively) may also help to protect the system from invasion by microbial competitors from the environment. Thus, kombucha can serve as a model system for addressing important questions about the evolution of cooperation and conflict in diverse multispecies systems. Further, it has the potential to be artificially selected to specialize it for particular human uses, including the development of antimicrobial ecosystems and novel materials. Finally, kombucha is easily-propagated, non-toxic, and inexpensive, making it an excellent system for scientific inquiry and citizen science.

Introduction

Kombucha is a traditional tea beverage fermented by a symbiotic community of acetic acid bacteria (AAB) (Acetobacteraceae) and osmophilic yeast (De Filippis et al., 2018). While the origins of the beverage are uncertain, records of the drink are found in early 19th century Russia (Dufresne & Farnworth, 2000). There is variation on the specifics of kombucha fermentation, but the typical process proceeds as follows: black or green tea is brewed for at least 5 min, supplemented with sucrose (5–10% (w/v)), cooled to room temperature (20 °C), and then inoculated with kombucha liquid (usually 10–20% (v/v)) from a previous batch (Jayabalan et al., 2014). A mature bacterial cellulose (BC) biofilm from a previously brewed kombucha culture (often called a “mother” or SCOBY, for Symbiotic Community of Bacteria and Yeast) is typically placed on top of the solution and allowed to ferment for 10–14 days. The presence of a carbon source in the solution, typically sucrose, initiates a cascade of metabolic processes that generates a carbonated and slightly acidic drink at the end of the primary fermentation cycle. One of the more striking aspects of the system is the floating cellulose pellicle that forms in tandem with fermentation; this biofilm is produced by the bacteria and encapsulates a microbial community within it (Marsh et al., 2014). The dominant bacterial genus in the system is Komagataeibacter (formerly Gluconacetobacter, and prior to that, Acetobacter, Yamada et al., 2012) (Marsh et al., 2014; Chakravorty et al., 2016) with numerous species identified within various kombucha cultures. These include Komagataeibacter xylinus (Reva et al., 2015; De Filippis et al., 2018), Komagataeibacter intermedius (Dos Santos et al., 2015; Reva et al., 2015; Gaggìa et al., 2019), Komagataeibacter rhaeticus (Machado et al., 2016; Semjonovs et al., 2017; Gaggìa et al., 2019), Komagataeibacter saccharivorans (Reva et al., 2015; De Filippis et al., 2018), and Komagataeibacter kombuchae (Reva et al., 2015). Another AAB genus often found in kombucha cultures is Gluconobacter (Reva et al., 2015; Chakravorty et al., 2016; Gaggìa et al., 2019). The yeast species in the system are even more variable, and can include yeast in the genera Zygosaccharomyces, Candida, Torulaspora, Pichia, Brettanomyces/Dekkera, Schizosaccharomyces, and Saccharomyces (Mayser et al., 1995; Teoh, Heard & Cox, 2004; Marsh et al., 2014; Jayabalan et al., 2014; Reva et al., 2015). The microbial profiles of kombucha seem to vary partly based on geographical origin (Mayser et al., 1995; Marsh et al., 2014), and the composition of the kombucha changes over time as it progresses through fermentation (Marsh et al., 2014). This process involves the enzymatic cleavage of sucrose and the subsequent processing of its monomer components into ethanol, acids, cellulose, and carbon dioxide (Jayabalan et al., 2014; Chakravorty et al., 2016). Brewed kombucha has antimicrobial properties that persist even with neutralization of the solution to pH 7 and thermal denaturation at 80 °C for 30 min (Sreeramulu, Zhu & Knol, 2000).

In this manuscript, we describe the potential uses of kombucha as a model system for studying cooperative and competitive interactions, and we also discuss the potential human uses of kombucha and kombucha-generated biofilms for nutrition, human health and industrial applications.

Survey methodology

This review was assembled via electronic searches on various platforms, including Google Scholar, Web of Science, and PubMed. In order to gain a background understanding of kombucha as a scientific model, searches were focused on the terminology adjacent to it and other fermented foods: “kombucha”, “kombucha tea,” “tea fungus,” “fermented tea,” “fermentation,” “SCOBY,” “biofilm,” “pellicle,” Examples of other fermented foods were examined as well, particularly those already used as model systems, such as sourdough and yogurt. Further development of the literature collection used keywords associated with microbial ecology and social interactions between microbes, such as: “cooperation,” “conflict,” “symbiosis,” “syntrophy,” “cheater.”

Searches evolved to include aspects of the four phases of development known to occur during kombucha fermentation: invertase production, ethanol fermentation, ethanol oxidation and acidification, and biofilm formation; many of these sources primarily used other organism models as references. Each of these sources provided a wealth of data far beyond the scope of this paper, and thus were culled down to focus on the microorganisms reported within the kombucha community. As the kombucha scientific literature remains in its infancy, resources frequently led outside of the core focus of the paper and thus were integrated only as required.

Microbial model systems help researchers address important theoretical and applied questions

There is a precedent for using model microbial systems for studying the evolution of cooperation, conflict and social behavior. Evolutionary biologists have used microbial systems to tackle ecology questions that would otherwise be unfeasible at a macroscale, such as exploring the selection pressures required for the evolution of cellular cooperation in the form of multicellularity (Strassmann, Zhu & Queller, 2000; Ratcliff et al., 2012), as well as adaptive radiation, diversification, and population fragmentation (Rainey & Travisano, 1998; Habets et al., 2006). Microbial model systems can be easily altered via genetic modifications and experimental evolution, and naturally produce many intriguing behavioral patterns—such as foraging, dispersal, collective assembly via biofilms, production of antagonistic chemicals, and quorum sensing (West et al., 2006), many of which resemble social processes among animals. Microbial systems have also been used to address phenomena that were originally developed to explain human interactions, such as the Prisoner’s Dilemma (Greig & Travisano, 2004) and the Tragedy of the Commons (MacLean, 2008).

However, most existing microbial model systems consist of systems with single species or only a few species. Artificial microbial systems can also be limited in their applicability to real world phenomena and their scalability to larger ecological questions (Jessup et al., 2004). This is in contrast to natural microbial systems, which are highly diverse and within which multispecies cooperation is likely to be complex. Despite advances in the ability of microbiologists to bring previously “unculturable” microbes into the lab (Vartoukian, Palmer & Wade, 2010) and the use of metagenomics and other “omics” approaches to analyze complex communities (Franzosa et al., 2015), the complexity of natural systems makes them difficult to reproduce and study in the laboratory (Wolfe & Dutton, 2015). We promote the idea that a compromise between natural and artificial systems can be found in fermented foods. They provide the ease of culturing microorganisms typical of simpler artificial systems, but include the diversity and complexity of natural systems. As a result, fermented foods and beverages may balance the advantages and disadvantages of natural and artificial systems.

Methods for propagating many fermented cultures are well-characterized due to their long history of cultivation and domestication. Genetic information exists for many of the microbes that are key to these fermentation processes and they have predictable cycles of development and succession, allowing for highly reproducible results despite their relative complexity (Wolfe & Dutton, 2015). During decades or centuries of domestication, artificial selection also likely favored fermented foods with cultures that are resistant to invasion by pathogenic species. The microbes in fermented foods produce factors that control the growth of potential invaders (Steinkraus, 1997) and thus help to stabilize the microbial population within the system. Researchers have found that kombucha does indeed have antimicrobial properties, including activity against many human pathogens (Greenwalt, Ledford & Steinkraus, 1998; Sreeramulu, Zhu & Knol, 2000). These features of kombucha make it a tractable model of interspecies dynamics that has implications for food preservation and for human health.

In this review, we provide a general overview of the cooperative and competitive interactions that occur during kombucha fermentation. For example, yeast produce the invertase enzyme which acts as a public good, breaking down sucrose that can then be used by both yeast and bacteria. The bacteria produce cellulose that becomes the pellicle, which may also act as a public good, protecting the liquid culture from colonization by competitors, delaying desiccation, and possibly acting as a resource store. All of these features of kombucha make it a promising model system for studying multispecies cooperation. Some of these characteristics point to other potential human uses for kombucha and kombucha-derived products, which we discuss at the end of the paper.

The social biochemistry of kombucha

Kombucha brewing begins with a solution of “sweet tea” (typically 5–10% (w/v) sucrose dissolved in brewed tea) and a small amount of kombucha starter culture (typically 10–20% liquid (v/v) and 2.5% biofilm (w/v)) from a previously fermented batch (Jayabalan et al., 2014). The teas used as substrates for kombucha are variable. Black and green teas are most commonly used, but are far from the only substrates tested (Jayabalan et al., 2014; Villarreal-Soto et al., 2018). Other substrates include oolong, jasmine, and mulberry teas (Talawat et al., 2006), rooibos (Gaggìa et al., 2019), coconut water (Watawana et al., 2016), and teas produced from various medicinal herbs (Battikh, Bakhrouf & Ammar, 2012; Velićanski, Cvetković & Markov, 2013).

Regardless of the initial substrate composition, the starter culture itself provides the main microbial inoculum into the solution. However, airborne or other environmental microbes may interact with or become part of the solution and potentially contribute to the microbial community. While the microbes ferment the substrate, enzymes produced by the yeast cleave sucrose into glucose and fructose and convert these monomers into ethanol and carbon dioxide. Next, bacterial enzymes oxidize ethanol, generating acetic acid that results in an low pH environment. The bacteria also produce cellulose which leads to biofilm formation (Jayabalan et al., 2014; Chakravorty et al., 2016) (see Fig. 1). Below we provide details about these metabolic processes and the microbial social interactions that occur over the course of kombucha fermentation (see Table 1).

Figure 1 Kombucha metabolism and microbial interactions.

(A) Kombucha is brewed by adding tea and table sugar to a small amount of kombucha starter which contains yeast and bacteria. These microbes begin to break down the sugar, leading to a metabolic cascade that ends with a bubbly, acidic and slightly alcoholic beverage. (B) During the process of fermentation, cooperative and competitive interactions occur among microbes. The production of the public good invertase by yeast, the removal of waste products through metabolization of alcohol and the generation of the cellulose pellicle by bacteria are potentially cooperative functions. Antimicrobial metabolites, low pH, and the generation of a physical barrier inhibit the growth of competitors.

Table 1 Over the course of kombucha fermentation, microbes cooperate and compete.

Many of these processes lead to products that have potential human uses as antiseptics and biomaterials.

Stage of fermentation	Competitive interactions	Cooperative interactions	Human uses	
Yeast produce invertase	Possible competition over invertase	Yeast producing invertase as a public good	Invert sugar, various fermentations	
Yeast ferment sugars into ethanol	Yeast inhibiting competitors with ethanol	Bacteria using ethanol as nutrient	Ethanol as an antiseptic and intoxicant	
Bacteria oxidize ethanol to produce acetic acid	Bacteria inhibiting competitors with acidification	Bacteria metabolizing ethanol as an energy source	Acid as an antiseptic	
Bacteria produce biofilm	Bacteria physically blocking competitors and creating anoxic environment in liquid	Spatially structuring kin, possible resource storage and protection from invading pathogens	Biomaterial, possibly one which protects from invasion by pathogens	

At the beginning of the kombucha fermentation process, yeast produce invertase which cleaves the disaccharide sucrose to its monosaccharide components, glucose and fructose. This phase appears to be the first opportunity for resource interaction between the microorganisms, as the freely liberated monomers are accessible to any microbe as a carbon source. Approximately 99% of the monosaccharides generated by Saccharomyces sp. invertase diffuse into the environment before the producing yeast can import them (Gore, Youk & Van Oudenaarden, 2009). Thus, neighboring cells receive the vast majority of the monomers produced by invertase secreted by a focal cell, suggesting that the production of invertase (and the resultant products) fits the classic definition of a non-excludable public good.

While the invertase produced by yeast appears to be a cooperative good, some yeast do not actually produce it (so called “cheaters”). Interestingly, a study with Saccharomyces cerevisiae has shown that yeast phenotypes that produce invertase are found in higher frequency than non-producer phenotypes when in the presence of Escherichia coli (Celiker & Gore, 2012). In other words, when yeast are grown in co-culture with bacteria, cooperative invertase-producing yeast outperform cheaters that do not produce invertase. Celiker & Gore (2012) suggest that the rapid depletion of resources by bacteria leads to a scarcity of sugar in the environment, which in turn increases the frequency of the invertase-producing strain (since they are able to capture about 1% of the sugars they produce, while non-invertase-producers are completely starved). Interestingly, during kombucha brewing, bacteria rapidly transform many of these sugars into the cellulose pellicle (see section below on resource storage). Thus it may be the case that bacteria—by removing sugars from the solution and putting them into the pellicle—change the selective pressures within the kombucha solution so as to favor invertase-producing strains of yeast.

After the yeast cleave sucrose into its component monomers using invertase, the yeast begin consuming these sugars and producing ethanol. Ethanol can be harmful to both yeast and bacteria, primarily via modifications to cellular membrane structure, function, and integrity (for a review on microbial tolerance to alcohols, see Liu & Qureshi, 2009). High levels of alcohol can threaten the viability of the microbes within kombucha. Potentially harmful levels of ethanol are reduced by bacteria that oxidize it and excrete acetic acid, thereby lowering the overall pH of the fermenting kombucha. These AAB that are part of the kombucha are obligate aerobes, meaning that they need access to oxygen to ferment (Saichana et al., 2015). In static conditions, production of the surface biofilm may increase access to oxygen for the microbes that are found within it, including yeast that are embedded in the cellulosic matrix. This may be another example of cooperation between yeast and bacteria that occurs during fermentation of the kombucha.

Some strains of Dekkera/Brettanomyces can also produce acetic acid in the presence of oxygen (Ciani & Ferraro, 1997); however, it is yet unclear what proportion of final acid is contributed by the yeast within kombucha. The dominant organic acids within the community are acetic acid, gluconic acid, and glucuronic acid (Jayabalan et al., 2014; De Filippis et al., 2018; Gaggìa et al., 2019), but additional acids have been detected and quantified; these include lactic acid (Jayabalan, Marimuthu & Swaminathan, 2007; Malbaša, Lončar & Djurić, 2008), citric acid (Jayabalan, Marimuthu & Swaminathan, 2007), malic acid, tartaric acid (Srihari & Satyanarayana, 2012) and a host of others (see Jayabalan et al. (2014) for a comprehensive list).

The variety and abundance of acids produced in kombucha raises the question: is the acid produced simply a waste product or does it provide some benefit for the microbes that produce it? In general, low pH in the solution can select for microbes that are tolerant to acid, while potential competitors and invaders are excluded or inhibited. Acid tolerant yeast in kombucha (such as Dekkera/Brettanomyces) are able to survive and even thrive within acidic conditions (Blomqvist, 2011; Steensels et al., 2015) that are deleterious to other yeast genera. Similarly, AAB that are present in kombucha (such as Komagataeibacter) are highly acid-tolerant, while other bacteria are far less tolerant and cannot survive in high acid conditions (Trček, Mira & Jarboe, 2015). The ability of the kombucha community to generate and tolerate these acidic conditions may provide an overall benefit in terms of protecting the system from invasion by competitor microbes. Indeed, kombucha has been found to be able to inhibit pathogens in vitro and part of this effect (though not all of it) has been ascribed to its acidic character (Greenwalt, Ledford & Steinkraus, 1998; Sreeramulu, Zhu & Knol, 2000).

The bacteria-produced biofilm might provide protection from invasion and allow resource storage

The most conspicuous facet of the kombucha symbiosis is the SCOBY (though the microbial community exists in the liquid culture as well as in the cellulosic biofilm). The biofilm initially forms a thin layer on the top of the liquid, as small bacteria-produced cellulose filaments rise to the top of the solution and aggregate together. The biofilm becomes larger and stronger with subsequent fermentations, often forming multiple pancake-like layers that are connected with filaments. Komagataeibacter xylinus, regarded as a model species for BC production (Ross, Mayer & Benziman, 1991), has been characterized as a core contributor to the cellulose production in some kombucha cultures (Reva et al., 2015; De Filippis et al., 2018) and has been previously described as the dominant species in regards to BC production in kombucha (Marsh et al., 2014). However, recent studies have illustrated an even wider diversity of Komagataeibacter spp. in kombucha than previously known. Interestingly, De Filippis et al. (2018) found that Komagataeibacter xylinus dominates during fermentation in green or black tea at 20 °C, while Komagataeibacter saccharivorans has a growth advantage at 30 °C. As previously mentioned, other members of this genus have also been identified in kombucha—these include Komagataeibacter intermedius and Komagataeibacter rhaeticus which were found to be abundant in green and black teas at 27 °C, while Gluconobacter entanii was identified nearly exclusively in kombucha fermented with rooibos teas (Gaggìa et al., 2019). From these studies, it is clear that the environment has an impact on the composition of community members; there is no apparent “canonical” species composition across all substrates and all culture conditions.

Accordingly, species of Acetobacter (Sievers et al., 1995; Chen & Liu, 2000; Dutta & Gachhui, 2006; Zhang, Zhang & Xin, 2011), Gluconacetobacter spp. (Yang et al., 2008; Trovatti et al., 2011) and Lactobacillus spp. (Wu, Gai & Ji, 2004; Zhang, Zhang & Xin, 2011) have also been found in kombucha cultures. For aerobic species that produce BC, such as Komagataeibacter xylinus, agitated bioreactors increase cell growth and cellulose yield by improving the oxygen transfer rate (Reiniati, Hrymak & Margaritis, 2017); however, rather than forming a surface pellicle, agitation produces spherical or asterisk-like particles of cellulose in the culture media (Bi et al., 2014; Singhsa, Narain & Manuspiya, 2018). The optimal dissolved oxygen concentration for BC yield for a strain of this bacteria was reported to be 10% in fed-batch cultures (Hwang et al., 1999), as greater oxygen concentrations result in a shift toward gluconic acid production and reduced cell viability, while lower concentrations reduce cell growth (Lee et al., 2014). Additionally, a variety of carbon sources have been tested and shown to influence the production of BC, with glucose, sucrose, fructose, mannitol, molasses, and various other organic wastes or extracts serving as substrates (for comprehensive reviews on species, strains, carbon sources, culture times, and cellulose yields, see Chawla et al., 2009; Shah et al., 2013; Jozala et al., 2016). While glucose is the ideal biosynthetic building block for cellulose production (Jozala et al., 2016), Komagataeibacter xylinus can convert it into gluconic acid via glucose dehydrogenase, which has a detrimental effect on cellulose production (Kuo et al., 2016). Interestingly, the addition of 1% (v/v) ethanol to a culture medium containing Gluconacetobacter hansenii inhibited cell growth—but resulted in an increase of BC production and a decline of non-cellulose producing mutants (Park, Jung & Park, 2003). It is unclear whether the yeast-produced ethanol in kombucha communities could perform a similar role with the bacterial partners, but this suggests an avenue for further exploration.

While the majority of focus has understandably been concentrated on the bacteria-produced biofilm, yeast are known to produce biofilms as well, particularly when a part of mixed species assemblages (Kawarai et al., 2007; Furukawa et al., 2010; León-Romero et al., 2016). It is entirely possible that yeast are contributing to the biofilm structure in kombucha as well. Dekkera/Brettanomyces have been shown to produce biofilms with surface adherence properties directly affected by pH and sugar concentrations (Joseph et al., 2007). These yeasts could also produce biofilms at different rates based on ploidy status (Ishchuk et al., 2016). These factors could account for some of the non-uniformity observed in the biofilms which grow as multiple layers with strands suspended down (see Fig. 2).

Figure 2 Typical appearance of kombucha biofilm.

At the top of the image is the multi-species biofilm which is made up of Komagataeibacter hansenii, Dekkera bruxellensis, Dekkera anomala, and Schizosaccharomyces pombe. Often, pendulous “strands” of material are seen dangling from the underside of the biofilm as they are in this image. The liquid underneath the biofilm is tea undergoing fermentation.

In addition to the physical thickness of the biofilm, the extracellular polymeric substances (EPS) of the matrix can inhibit the diffusion of antibiotics or invading cells (Stewart, 1996; Mah & O’Toole, 2001). The presence of this pellicle likely makes it more difficult for microbes landing on the surface to access free sugars that are within the kombucha solution. Williams & Cannon (1989) performed a battery of experiments to study the environmental role of the pellicle, including: using in vitro UV irradiation to show that the pellicle decreases the bacteria’s susceptibility to UV rays compared to non-pellicle controls, using apple slices as a substrate to show that the pellicle works to retain moisture in the environment, and that pellicle-forming bacteria strains are able to outgrow other, unspecified wild strains of bacteria and molds. Additional experimental work is needed to investigate whether the pellicle provides this protective function in kombucha.

Another possible benefit that the biofilm may provide is the storage of resources (Jefferson, 2004). The biofilm is made of EPS (produced by microbes) which acts as a reservoir of carbon (Flemming & Wingender, 2010). It can also include polysaccharides like levan which may function as a storage molecule (Limoli, Jones & Wozniak, 2015), as supported by studies on Pseudomonas syringae (Laue et al., 2006). This might allow it to function as a resource storage system that can only be accessed by the kombucha-associated bacteria and yeast inside the solution if/when sugars become unavailable (e.g., if the system is not being fed fresh, sugar-rich tea). However, further research is necessary to determine whether the pellicle is systematically broken down and used as a resource source during “starvation” of the kombucha. In addition—and related to this hypothesis—it could be that the removal of sugars from the solution by the bacteria creates an environment that favors cooperative invertase-producing yeast over cheater strains (see the section above on invertase; Celiker & Gore, 2012). In this way, the biofilm provides a resource storage function and also may modify the selective pressures on yeast, favoring cooperation. If the biofilm does indeed change the evolutionary dynamics within the system in ways that inhibit cheaters, this would be an intriguing parallel with certain processes that happen in multicellular bodies that encourage multicellular cooperation and inhibit cellular cheating (Aktipis et al., 2015).

Kombucha as a model system for studying cooperation

Kombucha is characterized by many different social processes including public good production and cooperation to exclude competitors. This makes it a useful model system for understanding the evolution of cooperation, both in general terms and more specifically in the context of cooperation in multispecies microbial systems. Some ostensibly mutually-beneficial relationships may instead have evolved as indirect exploitation of each partner’s waste products or represent by-products of otherwise selfish traits (West, Griffin & Gardner, 2007). There has also been discussion about whether strategies that permit the evolution of cooperation in social groups, such as cheater detection and cheater punishment, occur in microbial communities (Travisano & Velicer, 2004). Kombucha may be a good model system to test for cheater control systems in microbes and also to investigate the evolution of microbial traits that benefit other microbes.

Also, there is important work to be done to understand how multi-species cooperative systems evolve. Kombucha may be a tractable system in which to study this process. Theoretical models suggest that when cooperators get the benefits of interacting with one another (through a process called behavioral assortment), cooperation becomes more viable, regardless of whether individuals are related or even the same species (Fletcher & Doebeli, 2009). Behavioral assortment occurs when cooperators have more interactions with one another than with non-cooperators—and it is a general principle that can select for the evolution of altruism in diverse systems (Fletcher & Doebeli, 2009). Behavioral assortment is also at the heart of why strategies like “Walking Away” from non-cooperators selects for cooperation in both partnerships (Aktipis, 2004) and groups (Aktipis, 2011). Kombucha could provide a model system in which to experimentally test these models and uncover the mechanisms that influence the evolution of cooperation in multi-species systems.

A substantial body of work has focused on social interactions within biofilms, which run the gamut from cooperative to competitive (for a review, see Nadell, Drescher & Foster, 2016). As the vast majority of biological systems in the natural world exist as multi-species communities, it is important for biofilm researchers to use models that reflect this reality (Tan et al., 2017). The kombucha SCOBY offers an easily-reproducible platform to explore questions about synergy and antagonism in multispecies interactions. While advances in various “omics” technologies have allowed greater interrogation of such biofilms (Tan et al., 2017; Burmølle et al., 2014), technical challenges remain; these include tracking and maintaining various community members in mixed assemblages (Elias & Banin, 2012). We argue that the well-characterized organisms often found in the kombucha community could reduce the difficulties in such endeavors. Specifically, the substrates for cell growth are well-established (see previous sections on teas and carbon sources), there are selective media recipes designed to isolate various organisms in acidic or fermented conditions (Beuchat, 1993; Makdesi & Beuchat, 1996; Sharafi, Rasooli & Beheshti-Maal, 2010; Morneau, Zuehlke & Edwards, 2011), and the organisms are already well-adapted to in vitro-like conditions.

Kombucha can also provide a model system for investigating the evolution of cooperation among hosts and their microbes. There is growing interest in host-microbiome interactions (Rosenberg & Zilber-Rosenberg, 2016) and the evolutionary consequences of cooperation and conflict among hosts and their microbes (Alcock, Maley & Aktipis, 2014; Wasielewski, Alcock & Aktipis, 2016; Foster et al., 2017). Kombucha might provide a model system for certain aspects of the eukaryote-bacteria interactions that occur between hosts and their microbiomes. Both kombucha and host-microbiome interactions involve a close association between bacterial and eukaryotic cells. In kombucha, eukaryotes are represented by yeast genera, which can include Zygosaccharomyces, Candida, Torulaspora, Pichia, Brettanomyces/Dekkera, Schizosaccharomyces, and Saccharomyces (Mayser et al., 1995; Teoh, Heard & Cox, 2004; Marsh et al., 2014; Jayabalan et al., 2014; Reva et al., 2015). There is a long history of using yeast as a model system for human disease and health (Botstein, Chervitz & Cherry, 1997), which suggests that it may be viable to use kombucha as a model system for human health issues that involve interactions of the host eukaryotic cells with bacteria.

Fermented foods are diverse microbial ecosystems

Kombucha is not the only fermented food that is a potentially useful model system for studying multispecies cooperation. Cheese rinds are diverse biofilms formed by bacteria and fungi communities which are influenced by the various processing and aging steps associated with cheese production (Wolfe et al., 2014). Despite the collection of samples across 10 countries and 137 cheese types, Wolfe et al. (2014) show that these communities were dominated by 14 bacterial and 10 fungal genera and demonstrate highly reproducible successional dynamics across vast geographic distances. The adoption of cheese as a model system has produced numerous intriguing findings, such as widespread horizontal gene transfer in rind-associated bacteria (Bonham, Wolfe & Dutton, 2017) and fungi (Ropars et al., 2015), characterization of a suite of “domestication” genes in a common industrial milk fermenting bacterium (Passerini et al., 2010), the rapid experimental evolution of “domesticated” Penicillium mutants during serial propagation on in vitro cheese agar (Bodinaku et al., 2019), and the application of lactococcal-produced bacteriocins to influence the populations of starter bacteria and pathogenic contaminants (Guinane et al., 2005).

Another fermented food that has been established as a useful model system is sourdough. The production of sourdough involves an association between lactic acid bacteria and yeast using flour (typically wheat or rye) as a carbon and energy source (Gobbetti, 1998). Mature sourdough is the end product of a series of acidifying fermentation steps initiated by a diverse assemblage of lactic acid bacteria (both facultative and obligate heterofermentative as well as homofermentative species), aerobic gram-positive and Gram-negative bacteria, Enterobacteriaceae, yeasts, and molds—that eventually results in the dominance of a few obligate heterofermentative species of Lactobaccillus (sometimes Leuconostoc) and yeast (Minervini et al., 2014). Commonly isolated species of Lactobacillius bacteria include Lactobacilli sanfranciscensis, L. fermentum, L. plantarum, L. brevis, L. rossiae, and other members of the same genus (see Huys, Daniel & De Vuyst, 2013; Minervini et al., 2014), while yeast species are more diverse and commonly include Saccharomyces cerevisiae, Candida humilis, Kazachstania exigua, Pichia kudriavzevii, Wickerhamomyces anomalus, and Torulaspora delbrueckii (De Vuyst et al., 2016). The bacteria preferentially hydrolyze maltose and liberate glucose into the medium, allowing its use by neighboring microbes (particularly maltose-negative species of yeast) and stabilizing the cooperative interaction (De Vuyst & Neysens, 2005). Much like kombucha, the nature and quality of the substrate (flour for sourdough) and the fermenting conditions have a direct impact on the final community structure that develops, though the various methods used to sample, isolate, and identify the microbes preclude a conclusive link between a sourdough and its microbial consortia (De Vuyst et al., 2014). All of this variation affects the genera and species involved at various stages of fermentation, which in turn affects the final product (De Vuyst & Neysens, 2005).

Kefir may also be a useful fermented food for studying multispecies cooperation. It is a fermented milk beverage propagated by a starter “grain” composed of a complex but stable community of lactic acid bacteria, AAB, and yeast (Simova et al., 2002). As in kombucha and sourdough, the yeast component of the kefir community varies, with Saccharomyces and Candida as the genera most frequently identified, but it is also known to contain Kluyveromyces (Farnworth, 2005). The kefir system appears to be characterized by cooperation as well since yeast provide the bacteria with growth-promoting compounds (amino acids, vitamins) during the early stages of fermentation, and the resulting bacterial products can be exploited as a source of energy for the yeast (Loretan et al., 1998; Viljoen, 2001). These are just some of the fermented foods that allow opportunities for examining cooperation and competition in microbial communities—more generally, they may provide useful and tractable experimental systems for studying these social and evolutionary dynamics.

Kombucha and other fermented foods may provide benefits to humans

Fermented foods have many potential benefits to humans and so understanding the dynamics of cooperation and conflict among microbes in systems like kombucha can have important applications as well. In developing countries they already provide an important source of protein and vitamins (Steinkraus, 1997). The microbes in fermented foods also help protect the food from microorganisms that might otherwise spoil it—maintaining the nutritional quality of the food and helping to keep it safe for human consumption for long periods of time (Steinkraus, 1997). It is possible that kombucha may provide similar benefits. In addition, due to the ability of kombucha to inhibit pathogen growth via acidity (Greenwalt, Ledford & Steinkraus, 1998) and other mechanisms (Sreeramulu, Zhu & Knol, 2000; Bhattacharya et al., 2016; Shahbazi et al., 2018), kombucha and its constituents are excellent candidates for developing novel agents to control pathogens and food-spoilage microbes. This is a possibility we are currently exploring in our laboratory.

It is not yet known how kombucha influences human health, although tea polyphenols have been shown elsewhere to confer health benefits (Dufresne & Farnworth, 2000) including potentially decreasing cancer risk (Ann Beltz et al., 2006). During kombucha fermentation, it was shown that the concentration of polyphenols first decreases and then spikes at day 12, leading to higher levels of polyphenols than were originally in the solution, possibly due to the release of additional catechins or enzymes by cell lysis (Jayabalan, Marimuthu & Swaminathan, 2007). However, work by Gaggìa et al. (2019) has shown that polyphenol content increases at first and then decreases over further fermentation time. The type of tea and microbial population (and their interactions with each other) seem to directly influence the level of polyphenols. It is possible that these compounds in kombucha may have some positive effect on health, but more studies are needed to identify these potential influences on health and the mechanisms underlying them.

Cellulose biofilms produced by AAB—like the SCOBY in kombucha—have been developed into useful materials for medical and textile purposes. For example, cellulose biofilms have been developed for medical dressings (Lin et al., 2013), skin tissue repair (Fu, Zhang & Yang, 2013), incorporation into composite materials (Shah et al., 2013), and even clothing (Lee & Ghalachyan, 2015). These examples suggest that biofilms produced by kombucha fermentation can be used in a variety of beneficial products.

The role of viruses in fermented foods is currently unknown

An additional potential player in the cooperative and competitive dynamics inside kombucha and other fermented foods is viruses. The role of viruses in fermented foods has been largely unstudied due to challenges related to culturing, but recent advances have been made with the widespread adoption of metagenomic technologies (Park et al., 2011). Analysis of kimchi, sauerkraut, and fermented shrimp indicate that these foods contain less complex viral communities than environmental samples, possibly due to their limited microbial hosts (Park et al., 2011). Jung et al. (2011) found evidence that the phage burden of the dominant bacteria during the late stages of kimchi fermentation has a direct impact on bacterial abundance and accordingly affected the resulting community dynamics. In sauerkraut, the succession of host bacteria—and their subsequent effects on fermentation—may be directly influenced by the activity of their associated phages (Lu et al., 2003). Phages have not yet been studied in kombucha fermentation. But, given their presence in other fermented foods, it is likely that they play a role in kombucha.

Not all viruses are harmful—in fact, a growing body of research shows that many viruses may sometimes benefit their hosts (Roossinck, 2011, 2015). An intriguing topic for future study is the possibility that phages and other viruses maintain conditions that permit multispecies cooperation in kombucha and other fermented foods. Viruses have been shown to slow down microbial cell cycles (Nascimento, Costa & Parkhouse, 2012) and alter metabolism and the composition of metabolites that are produced (Sanchez & Lagunoff, 2015). These features may potentially stabilize microbial ecosystems during food fermentation, perhaps inhibiting the proliferation of some microbes while allowing others. Some viruses also have the capacity to restore intestinal morphology and mucosal immunity of germfree or antibiotic-treated mice without causing disease (Kernbauer, Ding & Cadwell, 2014), suggesting that they may also have a positive effect on host epithelial cells during digestion. It is possible that viruses contribute to the stability and viability of the ecological systems of fermented foods and the fermentation processes in the gut microbiome. To the extent that viruses stabilize microbial communities, their role in maintaining a microbial ecosystem favorable for viral growth may provide an evolutionarily beneficial strategy for those viral strains. The role of viruses in multispecies cooperative interactions has been underexplored and is worthy of future research.

Conclusions

Kombucha is a fermented tea that is brewed by combining sweet tea with a small amount of kombucha starter which contains both yeast and bacteria. Over the course of fermentation, the yeast and bacteria cooperate in many ways—some of which are known and some which require further characterization—to metabolize resources and keep out invading microbes. Kombucha offers a unique opportunity for exploring general questions about the evolution of cooperation and also for exploring more specific questions about cooperation in complex multispecies systems. The social lives of the microbes within the community—particularly how they exchange resources, signal potential partners, or even deter so-called “free-riders”—are exciting directions for future work. There are also many potential applications of kombucha for human nutrition, material development, and even for controlling the growth of harmful microbes.

The authors wish to thank the Microbiome and Behavior Project members and the Cooperation and Conflict lab for discussion and insights during the development of this article.

Additional Information and Declarations

Competing Interests

Author Contributions

Data Availability

The authors declare that they have no competing interests.

Alexander May prepared figures and/or tables, authored or reviewed drafts of the paper, approved the final draft.

Shrinath Narayanan authored or reviewed drafts of the paper, approved the final draft.

Joe Alcock approved the final draft, extensive discussions and emails.

Arvind Varsani contributed reagents/materials/analysis tools, authored or reviewed drafts of the paper, approved the final draft.

Carlo Maley contributed reagents/materials/analysis tools, authored or reviewed drafts of the paper, approved the final draft.

Athena Aktipis contributed reagents/materials/analysis tools, prepared figures and/or tables, authored or reviewed drafts of the paper, approved the final draft.

The following information was supplied regarding data availability:

The research in this article did not generate any data or code as it was collected from the literature and represents a review of existing material synthesized into a new context.

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
