# Peer review of "Kombucha: a novel model system for cooperation and conflict in a complex multi-species microbial ecosystem"

_PeerJ, doi:10.7717/peerj.7565_

## Round 0.1 · original submission · Major Revisions

Dear Dr. May and colleagues:

Thanks for submitting your manuscript to PeerJ. I have now received three independent reviews of your work, and as you will see, the reviewers raised some concerns about the manuscript. Despite this, these reviewers are optimistic about your work and the potential impact it will have on research studying Kombucha microbial ecology and dynamics. Thus, I encourage you to revise your manuscript accordingly, taking into account all of the concerns raised by the three reviewers.

Overall, the reviewers found the work to be a bit superficial, especially in regard to other similar reviews. It appears that relevant work is not cited, yet should be in such a review. The reviewers provide many examples. They also provide valuable suggestions on how to better support some your claims.

Importantly, please ensure that an English expert has proofed your revised manuscript.

I look forward to seeing your revision, and thanks again for submitting your work to PeerJ.

Good luck with your revision,

-joe

·

Basic reporting

There are many inaccuracy in microbiological and chemical hypotesis reported. References needs to be improved with more specific and recent literature . Moreover I think your English language should be more clear to make more comprensible at a greater audience.

Experimental design

1- Lane 46 - I suggest to delete “ethanol-producing” because the yeasts genus isolated in Kombucha’s samples is not specifically ethanol-producer
2- Lane 55-62 - Many recent papers De Filippis et al. 2018; Gaggia et al. 2019 described better the Kombucha’s microbiome. I suggest you to use more recent references.
3- Lane 82 - Change with yogurt.
4- Lanes 115-116 – Please support your thesis with references.
5- Lanes 130-132 - Why do you mention only lactic acid as control compound? Lactic acid is not the most representative control agent in fermented foods. Your thesis is not supported by appropriate references. I suggest you to correct and improve the sentence with more exact information supported by references.
6- Lane 145 - Please be more specific on the assertion “solution of sweet tea”. Describe the different sugar concentration extensively described in literature.
7- Lane 147 - Please describe the different substrates used in Kombucha fermentation supported by appropriate references.
8- Lanes 212-217 - Please describe the different organic acids entrust in the acidification process supported by recent references.
9- Lane 227-229 – For my opinion this is and incorrect information. Please change and improve the affirmation with more recent data based on 16s fingerprinting and NGS sequencing methods .I suggest to improve references.
10- Lane 240–253 - Hypotesis incorrect supported by insufficient references. Recent study described how oxygen and sugar concentrations are strictly linked to cellulose production in Komagataeibacter spp.
11- Lanes 268-272 - Unsopported hypothesis. Different autors showed how cellulose production in Kombucha increases during fermentation time.
12- Lane 301 - What do you mean “can favor”?
13- Lanes 338-343 - Incorrect assertion, many authors investigated the sourdough microbiome and recent studies shows the big difference between microbial ecology isolated in different sourdoughs.
14- Lanes 369-372 - Please improve with recent references. I suggest to improve the hypothesis illustrating the nutraceutical effect of organic acids typical of Kombucha’s beverage.
15- Lanes 377-378- Are you sure of this hypothesis? Gaggia et al. 2019 and other authors observed different dynamics.

Validity of the findings

The goals described in lanes 72-73 isn’t well developed in the conclusions and, in general in the study there are many argument that should be improved with appropiate literature

Additional comments

The study should be improved with more recent and relevant literature and I suggest to correct some microbial and chemical data incorrects.

Reviewer 2 ·

Basic reporting

The authors descibed the kombucha fermentation with microbiological details in a clear way. The article is original with very interesting outputs regarding the social behaviour of microorganisms. However, I found that mentioned literatures misses some recent papers on kombucha
(Marsh, A.J.; O’Sullivan, O.; Hill, C.; Ross, R.P.; Cotter, P.D. Sequence-based analysis of the bacterial and fungal compositions of multiple Kombucha (tea fungus) samples. Food Microbiol. 2014, 8, 171–178;
De Filippis, F.; Troise, A.D.; Vitaglione, P.; Ercolini, D. Different temperatures select distinctive acetic acid bacteria species and promotes organic acids production during Kombucha tea fermentation. Food Microbiol. 2018, 73, 11–16;
Gaggìa et al., 2019 Kombucha Beverage from Green, Black and Rooibos Teas: A Comparative Study Looking at Microbiology, Chemistry and Antioxidant Activity. Nutrients, 2019, 11(1), 1; doi:10.3390/nu11010001).

Experimental design

The manuscripts is within the aim and the scope of the journal and the topic is deeply investigated

Validity of the findings

The overall concept has its appropriate conclusion, very interesting, even if as pure microbiologist it was very hard to understand the phylosophy beyond.

Additional comments

58-59: this statement need a revision, since species within Acetobacteraceae are variable, also depending from the “mother” origin. (as the authors stated in lines 62-63 etc.)

82: please correct “yogourt”

124: please write “subjected”

229-230: authors did not mention others species belonging to Komagateibacter spp. that is the dominant genera in kombucha beverage; authors should also mention more recent paper such as
Marsh, A.J.; O’Sullivan, O.; Hill, C.; Ross, R.P.; Cotter, P.D. Sequence-based analysis of the bacterial and fungal compositions of multiple Kombucha (tea fungus) samples. Food Microbiol. 2014, 8, 171–178;
De Filippis, F.; Troise, A.D.; Vitaglione, P.; Ercolini, D. Different temperatures select distinctive acetic acid bacteria species and promotes organic acids production during Kombucha tea fermentation. Food Microbiol. 2018, 73, 11–16;
Gaggìa et al., 2019 Kombucha Beverage from Green, Black and Rooibos Teas: A Comparative Study Looking at Microbiology, Chemistry and Antioxidant Activity. Nutrients, 2019, 11(1), 1; doi:10.3390/nu11010001.
243-253 please revise the grammar; the colon should be replaced by semi-colon.

·

Basic reporting

see below

Experimental design

see below

Validity of the findings

see below

Additional comments

I am myself a big proponent of using fermented foods as model systems to study the ecology and evolution of microbial communities. I fully agree with the authors that these are excellent model systems for this purpose---including to study cooperation and conflict among microbes, as the authors propose.

Given the title and introduction of the review that is clearly focused on kombucha, the manuscript is for my taste too superficial with respect to the kombucha microbiology, ecology, and chemistry. I am myself not an expert on the kombucha system, but in comparison to previous literature, for example, the review of Jayabalan et al. (2014) Comprehens Rev in Food Sci and Food Safety, the present manuscript reads as somewhat of a superficial scan of the potential of the kombucha system. Additionally, in the places where the authors do introduce potential new points-of-view or ways in which the kombucha system could be utilised, the motivation for these ideas are often anecdotal and informal, rather than informed by the previous literature on the kombucha system. Some examples are:

- line 237: "the strands which are suspended from the pellicle and connect multiple layers of the pellicle have an appearance that suggest that they may partially be composed of yeast filaments (see Fig. 2)."

- line 256: "The pellicle forms on the surface of the liquid, and we have observed it making a tight junction with the sides of the container in which it is being grown."

- line 268: "Our informal observations of the kombucha systems indicate that the pellicle begins to degrade after being deprived of sweetened tea for several weeks,"

There are some more examples, but overall the reader (at least me) is left with the impression that the bulk of the author's knowledge is from anecdotal superficial observations of the kombucha system. I am sure that there is not the case, and hopefully by adding more detail with respect to cooperation and conflict in kombucha in particular can alleviate this impression.

Overall, I would like to again stress that I fully share the author's enthusiasm for using fermented foods to investigate interactions between microbes, but I am underwhelmed by the specifics of the kombucha system that I learn from the current manuscript compared to what could be applied to many other non-specific mixed-species microbial communities.


Minor comments:

line 178: "cultivation of bacteria by the yeast". This is a fun analogy, but I think there is (yet) no evidence to even suggest that the primary function of the bacteria storing resources in the pellicle is to cultivate yeast. Likely the bacteria would also perform this behaviour in absence of yeast. I of course can be wrong, but this should be formulated much more carefully.

---

## Round 0.2 · Minor Revisions

Dear Dr. May and colleagues:

Thanks for revising your manuscript. The reviewers are very satisfied with your revision (as am I). Great! However, per reviewer 3, there are a few minor edits to make. Please address these ASAP so we may move towards acceptance of your work.

Best,

-joe

·

Basic reporting

The autors corrected the manuscript extensively and I think they did a good research work

Experimental design

All the source are adeguadetely cited except:
- Gaggia et al. 2018 (correct with Gaggia et al. 2019)

Validity of the findings

For my opinion the autors developed a good research work

Reviewer 2 ·

Basic reporting

the authors have reviewed the manuscript according to the suggestions.

Experimental design

the authors have reviewed the manuscript according to the suggestions.

Validity of the findings

the authors have reviewed the manuscript according to the suggestions.

·

Basic reporting

The authors have made a considerable effort to add depth to their review, and as a result the review sits on a much stronger foundation.

Here are some final remarks:

- line 242: the "enzymes" don't ferment the sugar, right?

- line 261: "neighbouring cells receive the vast majority of sugars," ... of the monomers produced by invertase secreted by a focal cell.

- line 266: "certain bacteria" => directly specify E. coli, unless there are other examples.

- line 369: "environment has a massive impact" => it is rather suggested that the tea type has an impact, whether it is massive or not and why remains unclear.

- line 373: "species of Acetobacter spp..." => remove 'spp.', otherwise it reads "species of ... species"

- line 558: "as they have not had prolonged selection in the same environment as the native community, and thus are less likely to have the enzymatic toolkit required to break down the carbon polymers as readily." => I don't follow why there would be selection for using EPS in kombucha-associated bacteria but not **any** others. I find this argument rather weak.

- line 680: sourdough bacteria are not restricted to obligate heterofermentatitve species, but also include homofermentative and facultative heterofermentative species.

Experimental design

The survey appears exhaustive.

Validity of the findings

The title and aim of the manuscript puts the main emphasis on "cooperation and conflict". In this regard, I have to admit that the final sections on health benefits and viruses seem somewhat out of place. While I think the authors should maintain a high degree of "creative freedom" when writing reviews, this might be something that they would want to consider.

Additional comments

The language has suffered a bit in this revision. I would suggest to the authors to give the manuscript another read over to improve the flow.

---

## Round 0.3 · accepted · Accept

Dear Dr. May and colleagues:

Thanks for revising your manuscript to PeerJ, and for addressing the concerns raised by the reviewers. I now believe that your manuscript is suitable for publication. Congratulations! I look forward to seeing this work in print, and I anticipate it being an important resource for research on the microbial ecology of Kombucha.

Thanks again for choosing PeerJ to publish such important work.

-joe